# Efficacy of surgical management for recurrent intrahepatic cholangiocarcinoma: A multi-institutional study by the Okayama Study Group of HBP surgery

Toru Kojima[1☯], Yuzo Umeda[2☯]*, Tomokazu Fuji[1], Takefumi Niguma[1], Daisuke Sato[3], Yoshikatsu Endo[4], Kenta Sui[5], Masaru Inagaki[6], Masahiro Oishi[7], Tetsuya Ota[8], Katsuyoshi Hioki[9], Tadakazu Matsuda[10], Hideki Aoki[11], Ryuji Hirai[12], Masashi Kimura[13], Takahito Yagi[2], Toshiyoshi Fujiwara[2]

1 Department of Surgery, Okayama Saiseikai General Hospital, Okayama, Japan, 2 Department of Gastroenterological Surgery, Okayama University, Okayama, Japan, 3 Department of surgery, Hiroshima City Hiroshima Citizens Hospital, Hiroshima, Japan, 4 Department of Surgery, Himeji Japanese Red Cross Hospital, Hyogo, Japan, 5 Department of Gastroenterological Surgery at Kochi Health Sciences Center, Kochi, Japan, 6 Department of Surgery, National Hospital Organization Fukuyama Medical Center, Hiroshima, Japan, 7 Department of Surgery, Tottori Municipal Hospital, Tottori, Japan, 8 Department of Surgery, National Hospital Organization Okayama Medical Center, Okayama, Japan, 9 Department of Surgery, Fukuyama City Hospital, Hiroshima, Japan, 10 Department of Surgery, Tenwakai Matsuda Hospital, Okayama, Japan, 11 Department of Surgery, National Hospital Organization Iwakuni Medical Center, Yamaguchi, Japan, 12 Department of Surgery, Himeji Saint Mary's Hospital, Hyogo, Japan, 13 Department of Surgery, Matsuyama City Hospital, Ehime, Japan

☯ These authors contributed equally to this work.
* y.umeda@d9.dion.ne.jp

**Data Availability Statement:** All relevant data are within the manuscript and its Supporting Information files.

## Abstract

### Background

The prognosis of intrahepatic cholangiocarcinoma (ICC) has been poor, because of the high recurrence rate even after curative surgery. This study aimed to evaluate the prognostic impact of surgical resection of recurrent ICC.

### Patients and methods

A total of 345 cases of ICC who underwent hepatectomy with curative intent in 17 institutions were retrospectively analyzed, focusing on recurrence patterns and treatment modalities for recurrent ICC.

### Results

Median survival time and overall 5-year recurrence-free survival rate were 17.8 months and 28.5%, respectively. Recurrences (n = 223) were classified as early (recurrence at ≤1 year, n = 131) or late (recurrence at >1 year, n = 92). Median survival time was poorer for early recurrence (16.3 months) than for late recurrence (47.7 months, p<0.0001). Treatment modalities for recurrence comprised surgical resection (n = 28), non-surgical treatment (n = 134), and best supportive care (BSC) (n = 61). Median and overall 1-/5-year survival rates

**Funding:** Financial support was received from the Japan Society for the Promotion of Science (grant number 19K09217 to Yuzo UMEDA).

**Competing interests:** The authors have declared that no competing interests exist.

after recurrence were 39.5 months and 84.6%/36.3% for surgical resection, 14.3 months and 62.5%/2.9% for non-surgical treatment, and 3 months and 4.8%/0% for BSC, respectively (*p*<0.0001). Multivariate analysis identified early recurrence, simultaneous intra- and extrahepatic recurrence, and surgical resection of recurrence as significant prognostic factors. In subgroup analyses, surgical resection may have positive prognostic impacts on intra- and extrahepatic recurrences, and even on early recurrence. However, simultaneous intra- and extrahepatic recurrence may not see any survival benefit from surgical management.

## Conclusion

Surgical resection of recurrent ICC could improve survival after recurrence, especially for patients with intra- or extrahepatic recurrence as resectable oligo-metastases.

## Introduction

Intrahepatic cholangiocarcinoma (ICC) is the second most common primary malignant tumor of the liver, accounting for 10–20% of all primary liver malignancies [1]. Surgery has been regarded as a potentially curative treatment, providing ICC patients with a median overall survival (OS) of 14.4–38.8 months [2–4]. Unfortunately, many cases are diagnosed at an advanced stage, because ICC shows few specific early symptoms. Only about 20–40% of potentially operable patients are offered operative resection [5]. On the other hand, several reports have described the efficacy of systemic chemotherapy. While various regimens can achieve partial response, the effects seem limited [6–8].

Recurrence after curative surgery for ICC is common, with a reported recurrence rate of 50–79% [2, 4, 9, 10]. Many and various clinical factors have been identified as risk factors for recurrence and poor survival [11–15]. As with hepatocellular carcinoma (HCC), ICC shows a metastatic predilection for the liver, so locoregional therapy may represent a reasonable approach [16]. Ercolani et al. reported that aggressive multimodal treatment of recurrent ICC is associated with better outcomes [15]. A multi-institutional study showed that re-resection contributed to relatively better prognosis than systemic chemotherapy or best supportive care (BSC) [10]. A steadily improving understanding of risk factors for ICC recurrence and improved postoperative monitoring with modern imaging modalities is increasingly permitting diagnosis of recurrent ICC at an early stage while repeat resection is still technically feasible [5]. Considering these circumstances, re-evaluation of the efficacy of surgery for ICC recurrence appears worthwhile.

The aim of this study was to evaluate the prognostic impact of surgical resection for recurrent ICC, with a particular focus on the timing and patterns of recurrence.

## Materials and methods

### Study subjects

Participants in the present multicenter, retrospective comprised 404 adult subjects who had undergone hepatic resection with curative intent between January 2000 and December 2016. Clinical data for these subjects were collected from 17 medical institutions (Okayama University Hospital, Okayama Saiseikai General Hospital, Hiroshima Citizens Hospital, Kochi Health Sciences Center, Himeji Red Cross Hospital, National Fukuyama Medical Center,

Tottori Municipal Hospital, Tenwakai Matsuda Hospital, National Okayama Medical Center, Fukuyama City Hospital, Himeji St. Maria Hospital, Matsuyama Municipal Hospital, Sumitomo Besshi Hospital, Onomichi Municipal Hospital, National Iwakuni Medical Center, Himeji Central Hospital, and Kobe Red Cross Hospital). Of these, 12 institutions are board-certified training institutions for the Hepatobiliary and Pancreatic Surgery program in Japan [17]. Consequently, most patients were recruited from high-volume centers, leading to relatively standardized operative procedures and outcomes. Subjects meeting any of the following criteria were excluded: 1) insufficient clinical records (n = 35); 2) surgery-related death (n = 17); or 3) lack of follow-up data (n = 7). The definition of surgery-related death was mortality due to surgical complications within 90 days after surgery. On the other hand, comparatively early deaths due to recurrent tumor progression were not excluded. After excluding those individuals who met the exclusion criteria, a total of 345 subjects were included in this study.

The following demographic and clinical data were reviewed through medical records: age; sex; body mass index (BMI); history of diabetes mellitus; serum levels of carbohydrate antigen (CA)19-9 and carcinoembryonic antigen (CEA); maximum tumor diameter; number, localization, and morphology of tumors; surgical procedure; histological grade; vascular/serosal invasion; and timing and patterns of recurrence. With regard to localization of primary ICC, all ICCs were classified as hilar or peripheral type based on the anatomical origin of the tumor. The anatomical location of the tumor was judged from preoperative imaging such as computed tomography (CT) or magnetic resonance imaging (MRI). Tumors with the intrahepatic component and involvement of a large bile duct comparable with the intrahepatic second or third branches were defined as hilar type, whereas the other tumors involved in smaller than segmental branches were defined as peripheral type ICC.

## Follow-up protocol and diagnosis of recurrence

Patients with lymph node metastasis and/or positive surgical margins received adjuvant chemotherapy with regimens comprising gemcitabine and cisplatin or oral fluorinated pyrimidine for 6 months. After initial surgery, all patients were regularly followed-up every 3 months for the first 2 years, and every 6 months thereafter. At each visit, in addition to basic blood examinations, serum CA19-9 and CEA levels, contrast-enhanced chest and abdominal CT, and/or abdominal MRI were examined. Positron emission tomography (PET) was added in patients showing suspected subclinical recurrence or extrahepatic metastasis on CT or MRI. Diagnosis of recurrence was mainly based on these radiological findings with or without elevated concentrations of CA19-9. For cases without these definitive findings, diagnosis required endoscopic or percutaneous biopsy. Recurrence at ≤1 year postoperatively was defined as early recurrence, as reported previously [18]. Recurrence at >1 year was thus defined as late recurrence.

## Treatment modalities for recurrence and decision of them

Treatment strategy for each case of recurrence was assessed by a multidisciplinary team comprising liver surgeons, oncologists, hepatologists, and radiologists. Surgical resection for the recurrent site could be indicated, according to technical resectability, such as solitary or oligo-metastasis and patient conditions including performance status, estimated volume of future liver remnant, and feasibility and tolerability of repeat surgery. Of course, complete resection as R0 was required as the intent of repeat surgery. Patients who did not meet these criteria were treated by chemotherapy and/or radiation therapy as non-surgical treatment, or by BSC.

## Statistical analysis

Clinical variables were compared using the Mann-Whitney U test for continuous data and the Pearson's correlation coefficient for categorical data. Continuous variables are presented as median and interquartile range (IQR). Values of $p<0.05$ were considered significant. OS was evaluated using the Kaplan-Meier method and compared using log-rank testing. Cox's proportional hazard model was used to identify prognostic factors for recurrent cases. For this analysis, clinical variables showing values of $p<0.10$ in univariate analyses were entered into multivariate analysis. Hazard ratios (HRs) and 95% confidence intervals (95%CIs) were calculated. All statistical analyses were performed using JMP version 14 (SAS Institute Inc., Cary, NC, USA)).

## Ethics statement

This study conformed to the Declaration of Helsinki on Human Research Ethics standards and was approved by the Okayama University Hospital Institutional Ethics Board (number 1701–026). Since this study was retrospective in nature, there was no written informed consent from the investigated patients. All data were blinded before analysis.

## Results

Demographic characteristics of patients are provided in Table 1, and patient flow is summarized in Fig 1. The predominant tumor morphology was the mass-forming type (n = 256), followed by the periductal infiltrating type (n = 77), and intraductal growth type (n = 12). With regard to surgical procedures, more than 70% of patients underwent right/left hemi-hepatectomy or trisectionectomy as major hepatectomy, due to tumor extension. Most patients with tumor adjacent to the biliary confluence, such as hilar-type ICC, underwent bile duct resection. In our cohort, therapeutic lymph node dissection (LND) was performed for 235 patients (68%), of whom 96 patients showed positive lymph node metastasis on final histopathological examination; the rate of lymph node metastasis was 41%. The rate of positive surgical margins (including bile duct or liver cut surface) was 16%. Patients showing lymph node metastasis and/or positive surgical margins received adjuvant chemotherapy for 6 months.

A total of 223 patients showed recurrent ICC, with a median recurrence-free survival (RFS) of 17.8 months. Univariate analysis indicated the following significant risk factors for recurrence: preoperative CA19-9; maximum tumor diameter; periductal infiltrating type as the morphological type; multiple nodules; hilar-type ICC; requirement of major hepatectomy; LND; bile duct resection; vascular reconstruction; adjuvant chemotherapy; positive results for microscopic surgical margins; serosal invasion; positive lymph nodes; vascular invasion; and poorly/undifferentiated tumor (Table 1).

Patients with recurrence were divided into two groups according to the time to recurrence (TTR): early recurrence, ≤1 year after surgery (n = 131); and late recurrence, >1 year after surgery (n = 92). From other perspectives, pattern of recurrence was classified as intrahepatic only (n = 79); extrahepatic only (n = 109); or simultaneous intra- and extrahepatic recurrence (n = 35) (Fig 2a). The most frequent site of extrahepatic metastasis was lung (n = 49), followed by pleura/peritoneum including local recurrence (n = 42), lymph node (n = 40), bone (n = 11), and adrenal grand (n = 2) (Fig 2b). Treatment modalities for these recurrences comprised surgical resection (n = 28), non-surgical treatment (n = 134), and BSC (n = 61). Rates of surgical resection were high for intrahepatic-only recurrence and late recurrence. Surgical resection of recurrent sites comprised repeat hepatectomy for intrahepatic recurrence (n = 14) and local recurrence (n = 1), lung resection (n = 6), LND (n = 3), resection of local recurrence with bile duct (n = 1), adrenectomy (n = 1), and partial resection of abdominal wall (n = 1). On the

**Table 1.**

| Variables | All (n = 345) | Recurrence (n = 223) | No Recurrence (n = 122) | P-value[***] |
|---|---|---|---|---|
| **Parameters at initial resection** | | | | |
| Sex: Male / Female (%) | 214 (62%) / 131 (38%) | 138 (62%) / 85 (38%) | 76 (62%) / 46 (38%) | 0.94 |
| Age [*] | 70 (63–76) | 70 (63–76) | 69 (64–77) | 0.579 |
| BMI [*] | 22 (20–24.8) | 22 (19.9–24.5) | 22.2 (20.1–25.9) | 0.174 |
| Diabetes mellitus (%) | 66 (20%) | 41 (18%) | 25 (21%) | 0.634 |
| **Tumor factors** | | | | |
| CEA (ng/ml) [*] | 2.9 (1.8–5.8) | 2.9 (1.8–6.1) | 2.8 (1.8–9.9) | 0.394 |
| CA19-9 (U/ml) [*] | 39.8 (14.7–212) | 56.2 (16–456) | 27 (12.6–87.7) | 0.004 |
| Maximum tumor diameter (cm) [*] | 4.3 (2.8–6.7) | 4.8 (3.1–7) | 3.3 (2.4–5) | <0.0001 |
| Morphology | | | | |
| Mass forming / Periductal infiltrating / Intraductal growth (%) | 256 (74%) / 77 (22%) / 12 (4%) | 164 (74%) / 55 (25%) / 4 (2%) | 92 (75%) / 22 (18%) / 8 (7%) | 0.035 |
| Solitary / Multiple lesion (%) | 275 (80%) / 70 (20%) | 166 (74%) / 57 (26%) | 109 (89%) / 13 (11%) | 0.001 |
| Hilar type / Peripheral type (%) | 125 (36%) / 220 (64%) | 94 (42%) / 129 (58%) | 31 (25%) / 91 (75%) | 0.002 |
| **Treatment factors** | | | | |
| Major hepatectomy / Minor hepatectomy (%) | 247 (72%) / 98 (28%) | 170 (76%) / 53 (24%) | 77 (63%) / 45 (37%) | 0.009 |
| Lymph node dissection (%) | 235 (68%) | 161 (72%) | 74 (61%) | 0.027 |
| Bile duct resection (%) | 100 (29%) | 77 (35%) | 23 (19%) | 0.002 |
| Vascular reconstruction (%) | 26 (7.5%) | 21 (9%) | 5 (4%) | 0.073 |
| Adjuvant Chemotherapy (%) | 123 (35%) | 93 (42%) | 30 (25%) | 0.002 |
| **Pathological factors** | | | | |
| Microscopic surgical margin positive (%) | 56 (16%) | 45 (20%) | 11 (9%) | 0.007 |
| Lymph node metastasis (%) [**] | 96 (41%) | 79 (48%) | 17 (23%) | 0.0004 |
| Serosa invasion (%) | 139 (40%) | 111 (49%) | 28 (23%) | <0.0001 |
| Vascular invasion (%) | 219 (63%) | 165 (74%) | 54 (44%) | <0.0001 |
| fibrosis (%) | 90 (26%) | 52 (23%) | 38 (31%) | 0.113 |
| Poorly/undifferentiated (%) | 71 (21%) | 54 (24%) | 17 (14%) | 0.023 |

[*] Median and IQR: interquartile range,

[**] Among cases with lymph node dissection,

[***] Recurrence vs No Recurrence.

other hand, non-surgical treatment comprised systemic chemotherapy alone (n = 113), radiation (n = 10), chemo-radiation (n = 6), and radiofrequency ablation (n = 5). Clinical backgrounds of all recurrent cases according to treatment modalities are summarized in Table 2. In view of higher concentrations of preoperative CA19-9 and rates of lymph node metastasis, serosal invasion, and requirement of major hepatectomy, initially advanced ICC was significantly associated with non-surgical treatment or BSC. On the other hand, the group of patients who underwent surgical resection showed the highest induction rate of adjuvant chemotherapy (50%), followed by 48% for patients with non-surgical treatment, and 24% for patients receiving only BSC ($p$ = 0.006). In terms of timing and pattern of recurrence, early recurrences were less likely to receive intervention by surgical resection, compared with late recurrence ($p$ = 0.028). That is, early recurrences were treated using other modalities ($p$ = 0.028) (Fig 3). Intrahepatic- or extrahepatic-only metastasis in late recurrence could be selected for surgical resection: resection rates were 31% for intrahepatic recurrence alone, and 13% in extrahepatic recurrence alone. However, only 10% of simultaneous intra- and extrahepatic recurrences met the indications for surgical resection (Fig 1).

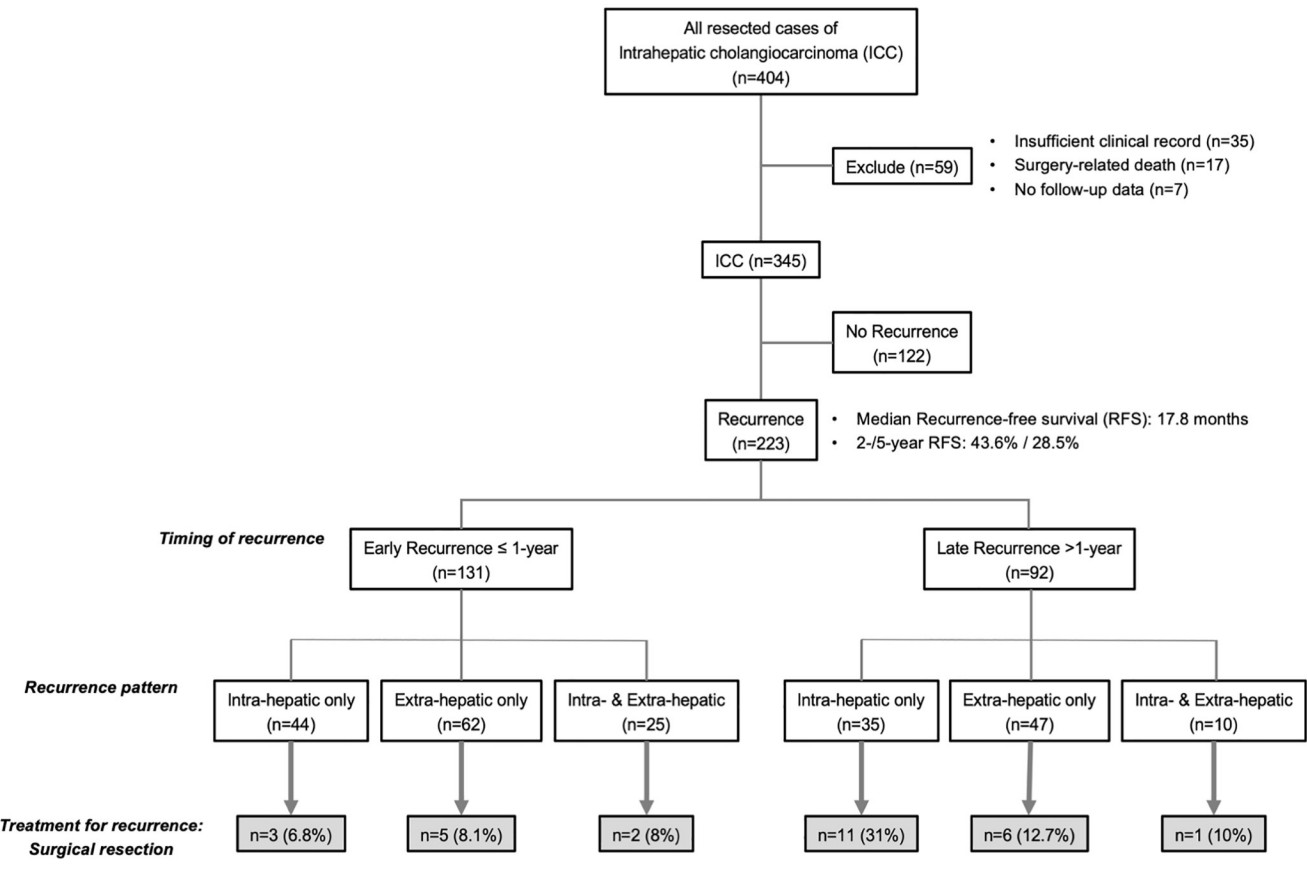

**Fig 1. Inclusion and exclusion criteria applied in the present study.**

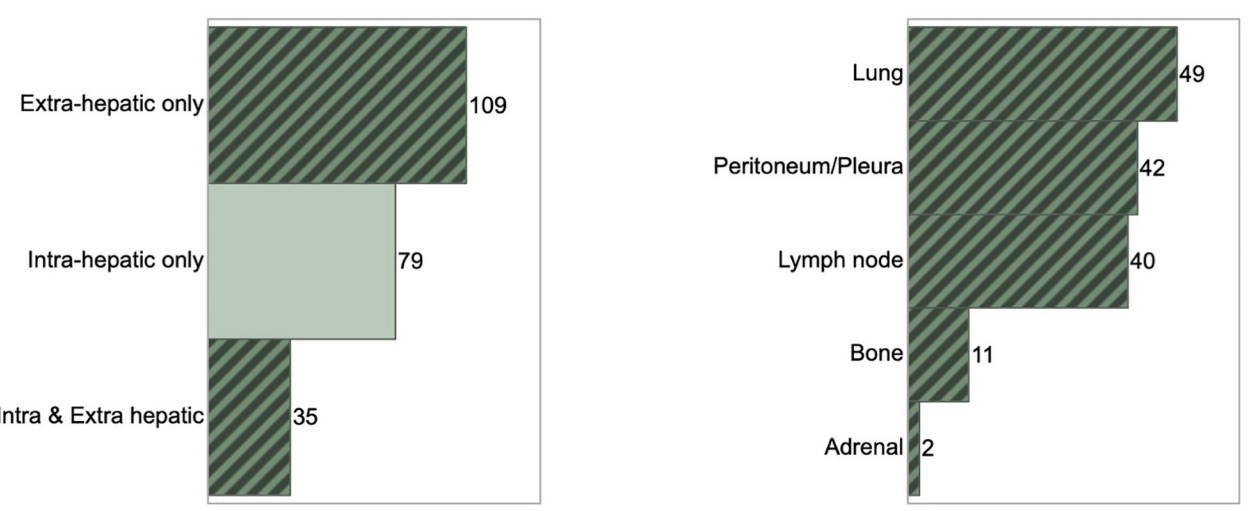

**Fig 2. Recurrent pattern (a) and sites of extrahepatic recurrence (b).**

**Table 2.**

| Variables | Surgical resection (n = 28) | Non-Surgical resection (n = 134) | BSC (n = 61) | P-value |
|---|---|---|---|---|
| **Parameters at initial resection** | | | | |
| Sex: Male / Female (%) | 16 (57%) / 12 (43%) | 82 (61%) / 52 (39%) | 40 (66%) / 21 (34%) | 0.724 |
| Age * | 71 (62–75) | 69 (62–75) | 72 (64–78) | 0.244 |
| BMI * | 23.4 (18.7–25) | 21.9 (20–24) | 22.7 (20.4–24.8) | 0.525 |
| Diabetes mellitus (%) | 5 (18%) | 28 (21%) | 8 (17%) | 0.794 |
| **Tumor factors** | | | | |
| CEA (ng/ml) * | 2.9 (1.5–4.3) | 2.8 (1.7–5.9) | 4.6 (2.1–10.1) | 0.053 |
| CA19-9 (U/ml) * | 18.3 (11–41) | 65 (16–439) | 146 (21.9–2030) | 0.004 |
| Maximum tumor diameter (cm) * | 4.9 (3.9–8) | 4.8 (3–7.2) | 4.8 (3–8.9) | 0.582 |
| Morphology | | | | |
| Mass forming / Periductal infiltrating / Intraductal growth (%) | 23 (82%) / 3 (11%) / 2 (7%) | 100 (75%) / 33 (25%) / 1 (0.5%) | 41 (67%) / 19 (31%) / 1 (1.5%) | 0.056 |
| Solitary / Multiple lesion (%) | 21 (75%) / 7 (25%) | 100 (75%) / 34 (25%) | 45 (74%) / 16 (26%) | 0.989 |
| Hilar type / Peripheral type (%) | 6 (21%) / 22 (79%) | 61 (45%) / 73 (54%) | 27 (44%) / 34 (56%) | 0.058 |
| **Treatment factors** | | | | |
| Major hepatectomy / Minor hepatectomy (%) | 18 (64%) / 10 (36%) | 99 (74%) / 35 (26%) | 53 (87%) / 8 (13%) | 0.04 |
| Lymph node dissection (%) | 17 (60%) | 101 (75%) | 43 (70%) | 0.272 |
| Bile duct resection (%) | 7 (25%) | 47 (35%) | 23 (37%) | 0.493 |
| Vascular reconstruction (%) | 2 (7%) | 12 (9%) | 7 (11%) | 0.776 |
| Adjuvant Chemotherapy (%) | 14 (50%) | 64 (48%) | 15 (24%) | 0.006 |
| **Pathological factors** | | | | |
| Microscopic surgical margin positive (%) | 7 (25%) | 27 (20%) | 11 (18%) | 0.748 |
| Lymph node metastasis (%)** | 6 (35%) | 43 (41%) | 30 (65%) | 0.016 |
| Serosa invasion (%) | 10 (35%) | 62 (46%) | 39 (64%) | 0.021 |
| Vascular invasion (%) | 19 (68%) | 102 (76%) | 44 (72%) | 0.614 |
| fibrosis (%) | 6 (21%) | 33 (25%) | 12 (21%) | 0.851 |
| Poorly/undifferentiated (%) | 8 (29%) | 30 (22%) | 16 (26%) | 0.716 |
| **Parameters at recurrence** | | | | |
| **Timing of recurrence** | | | | 0.028 |
| Early Recurrence ($\leq$ 1-year) | 10 (35%) | 82 (61%) | 39 (64%) | |
| Late Recurrence (>1-year) | 18 (64%) | 52 (38%) | 22 (36%) | |
| **Site of recurrence** | | | | 0.198 |
| Intrahepatic only | 14 (50%) | 50 (37%) | 15 (25%) | |
| Extrahepatic only | 11 (39%) | 63 (47%) | 35 (57%) | |
| Simultaneous Intra- & Extrahepatic | 3 (11%) | 21 (16%) | 11 (18%) | |

* Median and IQR: interquartile range,

** Among cases with lymph node dissection.

In survival analysis, 5-year RFS rate and median survival time (MST) were 28.5% and 17.8 months, respectively. Five-year OS rate and MST were 40.9% and 42.3 months, respectively (S1 Fig). MST and 1-/5-year OS after initial surgery were 140.6 months and 98.2%/89.2% in the no-recurrence group, 16.3 months and 61.5%/4.2% in the early recurrence group, and 47.7 months and 98.9%/37.9% in the late recurrence group, respectively (p<0.0001) (Fig 4a). As for survival after recurrence, simultaneous intra- and extrahepatic recurrence showed the shortest survival time, compared with extrahepatic-only and intrahepatic-only metastasis: MSTs were 9.4 months for simultaneous intra- and extrahepatic recurrence, 10.7 months for extrahepatic-

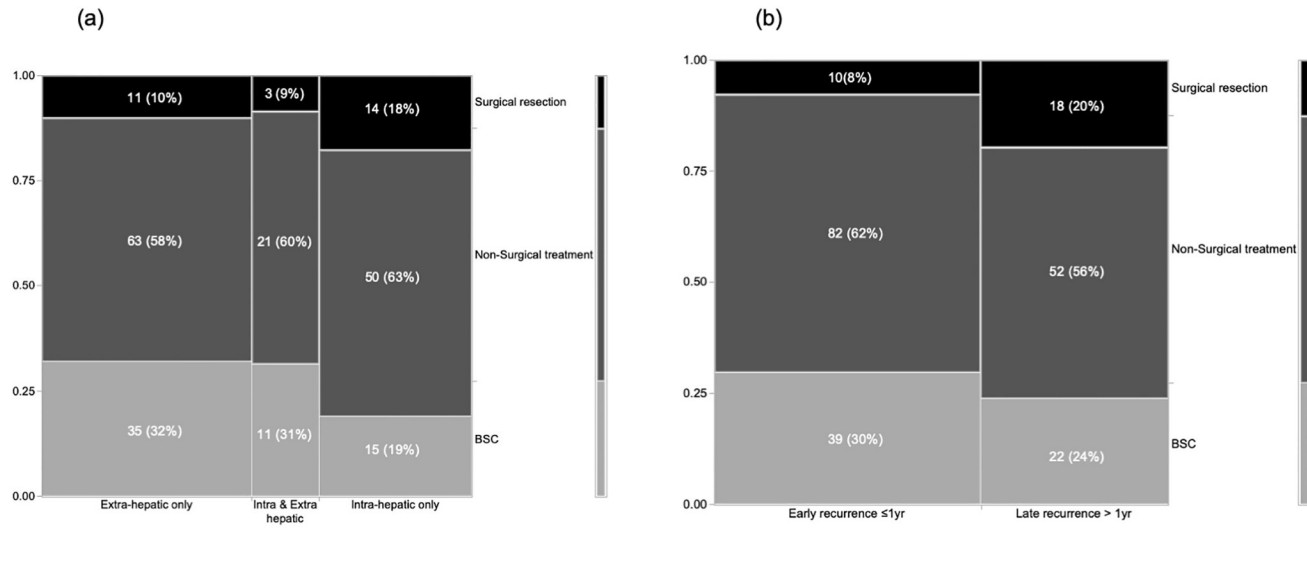

Pearson: p=0.198

Pearson: p=0.028

**Fig 3. Correlations between treatment modalities and sites (a) and timing (b) of recurrence.**

only recurrence, and 18.6 months for intrahepatic-only recurrence, respectively (p = 0.056) (Fig 4b). Regarding treatment modalities, surgical resection showed longer survival after recurrence (MST, 39.5 months) than non-surgical treatment (14.3 months; $p<0.0001$) or BSC (3.0 months; $p<0.0001$) (Fig 5a). In sub-group analysis according to recurrence pattern, survival benefit from surgical resection of the recurrent lesion was not recognized in patients with simultaneous intra- and extrahepatic metastasis, but was seen in intrahepatic- or extrahepatic-only metastasis (Fig 5b–5d). Furthermore, in the 223 cases with recurrence, Cox's proportional hazard modeling identified early recurrence (HR 1.39, $p$ = 0.046), simultaneous intra- and extrahepatic metastases compared with intrahepatic-only recurrence (HR 1.65, $p$ = 0.043), and surgical resection of recurrence compared with BSC (HR 0.06, $p<0.001$) or non-surgical treatment (HR 0.46, $p$ = 0.007) as independent prognostic factors for post-recurrence survival. In

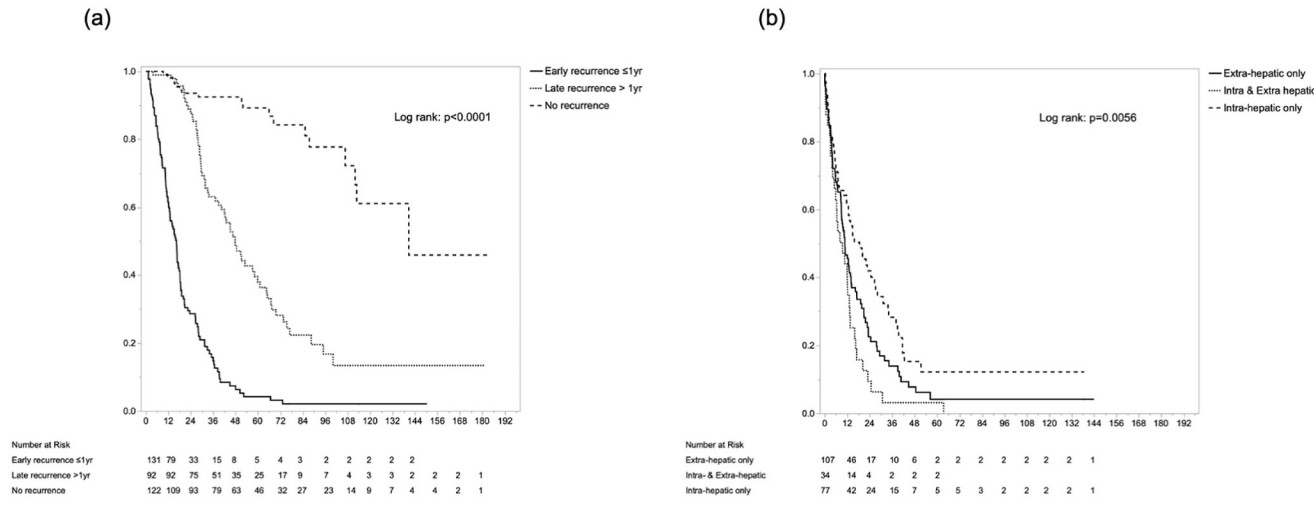

**Fig 4. Overall survival after primary resection, stratified by recurrence timing (a) and recurrence pattern (b).**

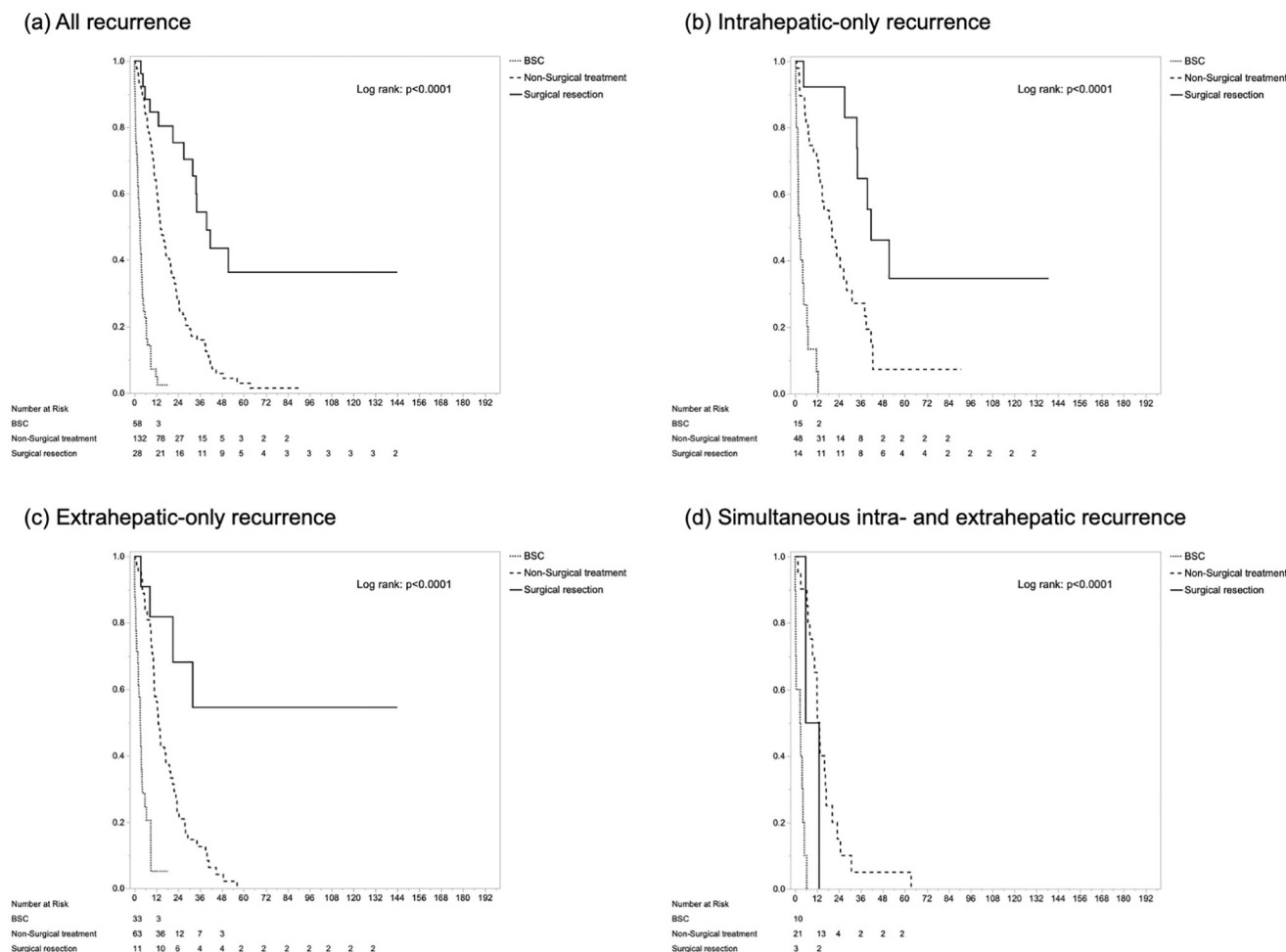

**Fig 5. Survival curves after recurrence stratified by treatment modalities in all patients with recurrence (a), intrahepatic-only recurrence (b), extrahepatic-only recurrence (c), and simultaneous intra- and extrahepatic recurrence (d).**

contrast, among factors at initial surgery, only hilar-type ICC (HR 1.60, *p* = 0.005) was selected as a significant factor (Table 3). Regardless of the timing of relapse, the superiority of surgical resection over other treatment modalities was evident (*p*<0.0001) (Fig 6a and 6b). Particularly in late recurrence, surgical resection resulted in long-term survival almost equivalent to that seen in no-recurrence cases.

## Discussion

In this study, 5-year overall RFS and MST were 28.5% and 17.8 months, respectively. In previous reports focusing on postoperative recurrence of ICC, median RFS has been reported as 11–17 months [5, 19–23]. Similar to those reports, our cohort showed recurrence approximately 1.5 year after initial surgery. Long-term recurrence and survival outcomes remain disappointing. Risk factors for recurrence after initial surgery are reported to include increased age, larger tumor diameter, macrovascular invasion, cirrhosis of the underlying liver, lymph node metastasis, and presence of multifocal disease. These are considered to be risk factors for both recurrence and poor survival [11–14]. Many reports have mentioned LND for ICC. The value of routine LND for ICC remains controversial [24–26]. Routine LND can facilitate

**Table 3.**

| Variables | | Univariate Analysis | | | Multivariate Analysis | | |
|---|---|---|---|---|---|---|---|
| | | Hazards ratio | 95% C.I. | P-value | Hazards ratio | 95% C.I. | P-value |
| **Parameters at initial resection** | | | | | | | |
| Sex: Male | vs Female | 1.220 | 0.927–1.608 | 0.154 | | | |
| Age | | 1.021 | 1.005–1.037 | 0.010 | 1.014 | 0.995–1.033 | 0.127 |
| BMI | | 1.012 | 0.974–1.052 | 0.520 | | | |
| Diabetes mellitus + | vs - | 0.859 | 0.605–1.220 | 0.396 | | | |
| **Tumor factors** | | | | | | | |
| CEA (ng/ml) | | 1.001 | 0.999–1.002 | 0.152 | | | |
| CA19-9 (U/ml) | | 1.000 | 0.999–1.000 | 0.120 | | | |
| Maximum tumor diameter (cm) | | 0.992 | 0.942–1.043 | 0.766 | | | |
| Macroscopic type | | | | | | | |
| Mass forming type | vs Periductal infiltrating type | 0.804 | 0.577–1.140 | 0.216 | | | |
| Mass forming type | vs Intra-ductal growth type | 1.414 | 0.594–4.616 | 0.471 | | | |
| Periductal infiltrating type | vs Intra-ductal growth type | 1.759 | 0.712–5.846 | 0.242 | | | |
| Multiple lesions | vs Solitary lesion | 1.096 | 0.809–1.485 | 0.551 | | | |
| Hilar type | vs Peripheral type | 1.440 | 1.095–1.895 | 0.009 | 1.602 | 1.154–2.223 | 0.005 |
| **Treatment factors** | | | | | | | |
| Major hepatectomy | vs Minor hepatectomy | 1.183 | 0.859–1.629 | 0.300 | | | |
| Lymph node dissection + | vs - | 0.977 | 0.723–1.321 | 0.881 | | | |
| Bile duct resection + | vs - | 1.241 | 0.942–1.648 | 0.122 | | | |
| Vascular reconstruction + | vs - | 1.043 | 0.663–1.642 | 0.853 | | | |
| Adjuvant Chemotherapy | vs no-Adjuvant Chemotherapy | 0.999 | 0.761–1.310 | 0.994 | | | |
| **Pathological factors** | | | | | | | |
| Microscopic surgical margin positive | vs negative | 1.103 | 0.793–1.534 | 0.558 | | | |
| Lymph node metastasis + | vs - | 1.363 | 1.001–1.858 | 0.048 | 1.032 | 0.741–1.437 | 0.850 |
| Serosa invasion + | vs - | 1.193 | 0.913–1.559 | 0.196 | | | |
| Vascular invasion + | vs - | 1.129 | 0.831–1.531 | 0.435 | | | |
| fibrosis + | vs - | 1.059 | 0.771–1.458 | 0.720 | | | |
| Poorly/undifferentiated | vs Well/mod. differentiated | 1.001 | 0.737–1.373 | 0.970 | | | |
| **Parameters at recurrence** | | | | | | | |
| Early Recurrence (≤ 1-year) | vs. Late recurrence (>1-year) | 1.501 | 1.145–1.981 | 0.003 | 1.398 | 1.005–1.946 | 0.046 |
| Recurrence site | | | | | | | |
| Simultaneous intra- & extrahepatic | vs. Intrahepatic only | 1.719 | 1.143–2.589 | 0.009 | 1.646 | 1.015–2.671 | 0.043 |
| Simultaneous intra- & extrahepatic | vs. Extrahepatic only | 1.278 | 0.864–1.879 | 0.220 | 1.387 | 0.880–2.184 | 0.157 |
| Extrahepatic only | vs. Intrahepatic only | 1.348 | 1.003–1.813 | 0.048 | 1.187 | 0.835–1.687 | 0.339 |
| Treatment for recurrence | | | | | | | |
| Surgical resection | vs BSC | 0.063 | 0.037–0.110 | <0.001 | 0.059 | 0.029–0.118 | <0.001 |
| Surgical resection | vs Non-surgical treatment | 0.439 | 0.287–0.684 | <0.0003 | 0.463 | 0.264–0.809 | 0.007 |
| Non-surgical treatment | vs BSC | 0.145 | 0.100–0.210 | <0.001 | 0.128 | 0.082–0.200 | <0.001 |

accurate staging with precise identification of nodal status, and can predict indication for adjuvant therapy [27]. Given these valuable aspects, LND of regional nodes may be considered as a standard option [28]. Adjuvant chemotherapy was unable to improve prognosis for all ICC patients after surgical resection, but could provide a potential survival benefit in subgroups of patients exhibiting increased risk, such as advanced tumors or positive lymph node metastasis [29, 30]. In our patient cohort, approximately 70% of patients showed positive nodal status. Interestingly, the induction rate of adjuvant chemotherapy was higher in the group with

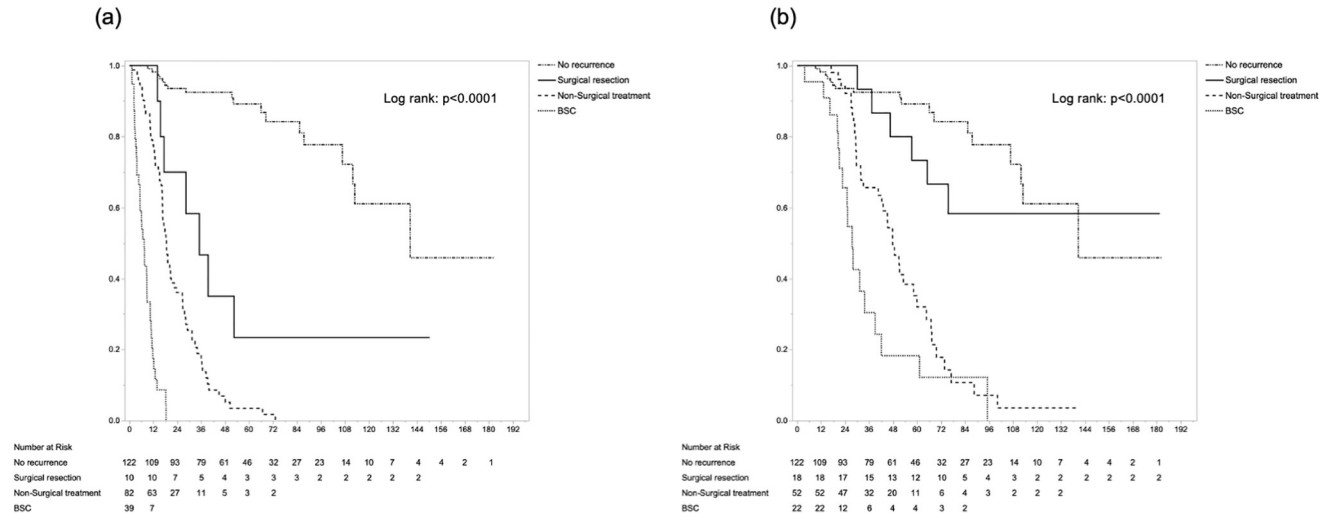

**Fig 6. Overall survival after primary resection by treatment modality, compared with no-recurrence patients, in early recurrence (≤1 year; a) and late recurrence (>1 year; b).**

surgical resection than in those with non-surgical treatment or BSC (p = 0.004). Perhaps this data suggested that adjuvant chemotherapy could increase the resection rate via control of cancer spread, leading to oligo-metastasis.

The rate of surgical resection seems to correlate with the timing and type of recurrence. In other words, this could be affected by the spread of cancer. Preoperative levels of CA19-9 in non-surgical treatment and BSC were thus higher than that with surgical resection. Of course, taking into account high rates of lymph node metastasis, serosal invasion, and requirement for major hepatectomy, initial advanced tumor would result in aggressive recurrence treated by non-surgical treatment or BSC, instead of surgical resection. In addition, preoperative CA19-9 could offer a promising predictive biomarker implying subclinical cancer spread at initial surgery.

No clear treatment guideline is currently available for recurrence, especially for patients with localized or systemic intrahepatic ICC recurrence. Several studies have evaluated the impact of various treatments, such as repeat hepatectomy, radiofrequency ablation, chemotherapy or radiotherapy on survival following recurrence of ICC [5, 15, 16, 19–22]. Although each report has shown the limitation of being a retrospective analysis, surgical resection of the site of recurrence was clearly established as an effective therapeutic option. In those reports, median post-recurrence survival after surgical resection was reported as 20–45 months. In this study, surgical resection of recurrent lesions showed 39.5 months as the MST and 84.6% and 36.3% as the 1- and 5-year OS rates after recurrence, significantly better than those from non-surgical treatment and BSC. Surgical resection can obviously provide clear survival benefits to patients with intrahepatic-only or extrahepatic-only recurrence. Conversely, surgical treatment may not be appropriate for simultaneous intra- and extrahepatic recurrence. Compared with previous reports, our study showed that post-recurrence survival seemed slightly better after surgical resection. However, these differences would be derived just from patient selection for surgical resection; in our patient cohort, the indication of surgical resection for recurrent ICC was limited to cases with the prospect of R0. Even though surgical resection is the best treatment modality for recurrent ICC, non-curative repeat surgery could end up providing outcomes just as dismal as those from non-surgical treatment [23]. This efficacious

treatment option thus should not be adopted for 'debulking' effects, but under a radical "cleaning-up" policy for R0.

With regard to tumor localization, hilar type ICC was indicated as one of significant prognostic factors. The pathological background and gross and histological features of ICCs are reported to differ according to the anatomical site [31, 32]. The hilar type ICCs originated from intrahepatic large biliary ducts are likely to show aggressive course with metastatic potential. Though tumor localization could be classified according to radiological findings in this study, classification based on radiological findings has been reported to accurately reflect histomorphological typing [33, 34]. In fact, hilar type ICC, defined by radiological localization in this study, showed higher recurrence rate and poorer survival than peripheral type. The low resection rate for recurrence in primary major hepatectomy cases and hilar type ICCs may be due to the grade of hilar type ICCs as well as the reduced liver reserve caused by the initial surgery, which may influence the decision to treat patients at the time of recurrence and the post-recurrent survival.

TTR has been reported as a crucial factor to predict prognosis after recurrence [5]. This interval is closely associated with tumor biology, including the metastatic potential of intra- or extrahepatic metastasis. In HCC, optimal cut-off values for differentiating between early and late tumor recurrence remain controversial [35]. Few studies have explored the issue in ICC, but TTR ≤ 1 year has been proposed as a valuable cut-off for early recurrence of ICC [18, 36]. According to those reports, we classified two groups to differentiate early and late recurrence. This cut-off was likely adequate, allowing could clear differentiation of survival both after initial surgery and after recurrence. Notably, surgical resection of recurrent ICC could show a positive prognostic impact even for early recurrence. Based on previous reports, surgical intervention for early recurrence showed poorer prognosis than that for late recurrence [5]. On the other hand, another report found that early recurrence did not affect post-recurrence survival [22]. Thus, regardless of TTR, surgical resection exerts a positive prognostic effect on survival after initial surgery or recurrence. However, in early recurrence, only patients with biologically low-grade recurrences or limited recurrent disease could benefit from surgical resection. In other words, the power of surgical resection relies heavily on the degree of recurrent-tumor distribution. Judging whether recurrent tumor represents limited disease is thus essential.

## Conclusions

Even if recurrences seem resectable, careful follow-up with chemotherapy may be advisable to determine biological malignancy. On the other hand, surgical resection for late-phase recurrence could provide a curative option offering equivalent prognosis to that of no-recurrence cases. While surgery remains the only way to obtain radical cure in ICC, surgery alone cannot achieve cure. Considering treatment strategies for ICC, initial surgery is only the first step and the introduction of adjuvant chemotherapy based on accurate staging should follow. In cases with recurrence, the path of surgical resection should always be explored to improve prognosis.

## Supporting information

**S1 Fig. Overall and recurrence-free survival curves.**
(TIFF)

## Acknowledgments

The authors thank their colleagues who contributed to data collection for this study: Kazuyasu KOBAYASHI (Sumitomo Besshi Hospital), Toshihiro MURATA (Onomichi Municipal

Hospital), Yasuhiko ISHIDA (Himeji Central Hospital), and Nobuhiro ISHIDO (Kobe Red Cross Hospital).

## Author Contributions

**Conceptualization:** Tomokazu Fuji.

**Data curation:** Toru Kojima, Tomokazu Fuji, Takefumi Niguma, Kenta Sui, Masaru Inagaki, Tetsuya Ota, Katsuyoshi Hioki, Tadakazu Matsuda, Hideki Aoki, Ryuji Hirai, Masashi Kimura.

**Formal analysis:** Yuzo Umeda.

**Funding acquisition:** Yuzo Umeda.

**Investigation:** Tomokazu Fuji, Daisuke Sato, Yoshikatsu Endo, Masahiro Oishi.

**Methodology:** Yuzo Umeda.

**Project administration:** Yuzo Umeda.

**Supervision:** Yuzo Umeda, Takahito Yagi, Toshiyoshi Fujiwara.

**Writing – original draft:** Toru Kojima, Yuzo Umeda.

**Writing – review & editing:** Yuzo Umeda.

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
