## [Decision Letter · Decision Letter 0]

17 Jul 2020

PONE-D-20-19960

Efficacy of surgical management for recurrent intrahepatic cholangiocarcinoma: A multi-institutional study by the Okayama study group of HBP surgery

PLOS ONE

Dear Dr. Yuzo Umeda,

Thank you for submitting your manuscript to PLOS ONE. After careful consideration, we feel that it has merit but does not fully meet PLOS ONE’s publication criteria as it currently stands. Therefore, we invite you to submit a revised version of the manuscript that addresses the points raised during the review process.  The Editorial team found merit in your study and we encourage a resubmission.

Please submit your revised manuscript within 60 days. If you will need more time than this to complete your revisions, please reply to this message or contact the journal office at plosone@plos.org. Please include the following items when submitting your revised manuscript:

We look forward to receiving your revised manuscript.

Kind regards,

Gianfranco D. Alpini

Academic Editor

PLOS ONE

Journal Requirements:

2. In ethics statement in the manuscript and in the online submission form, please provide additional information about the patient records/samples used in your retrospective study. Specifically, please ensure that you have discussed whether all data/samples were fully anonymized before you accessed them and/or whether the IRB or ethics committee waived the requirement for informed consent.

4. Please include your tables as part of your main manuscript and remove the individual files. Please note that supplementary tables (should remain/ be uploaded) as separate "supporting information" files

Reviewers' comments:

Reviewer's Responses to Questions

**Comments to the Author**

1. Is the manuscript technically sound, and do the data support the conclusions?

Reviewer #1: Yes

Reviewer #2: Yes

Reviewer #3: Partly

2. Has the statistical analysis been performed appropriately and rigorously? 

Reviewer #1: Yes

Reviewer #2: I Don't Know

Reviewer #3: No

3. Have the authors made all data underlying the findings in their manuscript fully available?

Reviewer #1: Yes

Reviewer #2: Yes

Reviewer #3: Yes

4. Is the manuscript presented in an intelligible fashion and written in standard English?

Reviewer #1: Yes

Reviewer #2: Yes

Reviewer #3: Yes

5. Review Comments to the Author

Reviewer #1: Toru KOJIMA et al aimed to evaluate the prognostic impact of surgical resection of recurrent ICC using a retrospective approach including 345 cases of ICC who underwent hepatectomy with curative intent in 17

institutions.

The study is of interest in a difficult cancer field with limited therapeutic options.

Authors highlight those who have a late, intrahepatic recurrence who are ideal candidate for surgery with acceptable outcomes.

I have only minor comments:

Abstract is confusing in the report of the survival and recurrence rate.

Table 1 is not properly formatted.

Reviewer #2: I read the manuscript by Kojima et al with great interest. The study is a multi center study with a large cohort of intrahepatic CCA.

I think the study is well performed and cohort number is big enough, However the tables are arranged in a strange style that it is VERY hard to follow. Therefore they need to re-do all tables. Figure are nice and clear.

Reviewer #3: This paper reported the result of retrospective study conducted in 17 surgical centers from Japan focused on recurrence patterns and treatment modalities for recurrent ICC. A total of 345 cases of ICC who underwent hepatectomy were retrospectively analyzed. Treatment modalities for recurrence comprised surgical resection in only 28 of the patients, non-surgical treatment in n=134, and best supportive care (BSC) in n=61.  In subgroup analyses, surgical resection may have positive prognostic impacts on intra and extrahepatic recurrences, and even on early recurrence.  Although the efforts of the authors are worthy of consideration, both the design of the study, both the limited number of patients which underwent surgery for recurrence, are serious limitations potentially associated with biased conclusions. 

The design of the study and the very low number (28 subjects; less than 10% of the resected subjects and average 12% of the recurrent subjects) are serious limitations of the study in view of the aim. The claim of the conclusion seems overstated with respect to the results and the evidence provided. Indeed, patients undergoing surgery for recurrence are likely to have been diagnosed in an early phase, may have better general condition which allow the surgery, and no advanced underlying liver disease, with respect to the non resected recurrent patients. Please analyze confounder factors which may have influenced the decision making. 

The current accepted classification of the CCA based on location identifies iCCA, perihilar CCA and distal CCA. The origin of the second order branches of the biliary tree define the iCCA vs pCCA. Thus it is not clear whether pCCA has been misclassified as iCCA in this study. Please explain.  

Please provide an accurate definition of the hilar-type ICC. 

Is the histomorphology subtype, small bile duct type vs large bile type, associated with the recurrence rate.

6. PLOS authors have the option to publish the peer review history of their article (what does this mean?). If published, this will include your full peer review and any attached files.

Reviewer #1: No

Reviewer #2: No

Reviewer #3: No

---

## [Author Response · Author response to Decision Letter 0]

5 Aug 2020

Aug 3, 2020

Gianfranco D. Alpini

Academic Editor

PLOS ONE

Manuscript ID: PONE-D-20-19960

Title: Efficacy of surgical management for recurrent intrahepatic cholangiocarcinoma: A multi-institutional study by the Okayama study group of HBP surgery

Authors: Kojima T et al.

Dear Dr. Alpini:

Thank you for your e-mail of July 17, 2020. We were pleased to know of your positive evaluation of our manuscript and its potential acceptance for publication in PLOS ONE, subject to adequate revision and response to the reviewers' comments. 

Based on your instructions, we uploaded the clean and marked files of the revised manuscript. We also pasted in the allocated space on the website our point-by-point responses to the comments raised by the reviewer. 

To allow easy access to our response to the comments raised by the reviewer, we also include a copy of our response in this letter. Essentially, we agreed with all the comments raised by the reviewer. 

We take this opportunity to thank the reviewer for the hard work. We also thank you for allowing us to resubmit a revised copy of the manuscript.

I hope that the revised manuscript is now acceptable for publication in PLOS ONE. 

Sincerely Yours,

Yuzo UMEDA, MD, PhD

Okayama University Graduate School of Medicine and Dentistry

Department of Gastroenterological Surgery 

Shikata-cho, Okayama-shi, Okayama 700-8558, JAPAN

E-mail: y.umeda@d9.dion.ne.jp

Point-by-point response to the comments of Reviewer 1

We thank the reviewer for evaluating our manuscript. The following text describes our responses to the comments made by the reviewer. 

1. The reviewer recommended a comprehensive statement about survival and recurrence rate in the Abstract. Accordingly, we changed the phrase “post-recurrence survival rates” to “survival rates after recurrence”. 

Page 3, line 10 in the Abstract section.

2. We apologize for our inadequate table content. We mounted the proper formatted table 1, 2, and 3.

Page 8, Table 1 was inserted as the proper format. 

Page 10-11, Table 2 was inserted as the proper format.

Page 12-13, Table 3 was inserted as the proper format.

Point-by-point response to the comments of Reviewer 2

We thank the reviewer for evaluating our manuscript. The following text describes our responses to the comments made by the reviewer. 

1. We apologize for our inadequate table contents. 

We modified all tables and submitted the proper formatted table 1, 2, and 3 in the revised document.

Page 8, Table 1 was inserted as the proper format. 

Page 10-11, Table 2 was inserted as the proper format.

Page 12-13, Table 3 was inserted as the proper format.

Point-by-point response to the comments of Reviewer 3

We thank the reviewer for evaluating our manuscript and greatly appreciated the reviewer's very important and worthwhile comments. The following text describes our responses to the comments made by the reviewer. The line number mentioned in the response to the comment refers to the small-size number that appears on the left margin of the text of the revised manuscript.

1. As noted by the reviewer, patients undergoing surgery for recurrence had a lower rate of Hilar type ICC and did not undergo major hepatectomy compared to non-resected recurrent patients, and this may have influenced the underlying remnant liver function in their decision-making. In results and discussion, these aspects were added.

Page 15, line 8-10 in the discussion section.

2. We appreciate the valuable comments, as the Reviewer pointed out ‘the definition of hilar/peripheral ICC’ is a very important issue. Firstly, the tumor involving hilar bile duct were not included. And classification of hilar/peripheral type was practically based on preoperative imaging. So we provide the accurate definition of hilar/peripheral type ICC in this study. 

Page 5, line 20 to Page 6 line 2 in the materials and methods section.

3. We agree with the prognostic impact of histomorphological subtypes. Unfortunately, due to the limitation of multicenter retrospective study, it was not possible to identify these subtypes in all resected cases. 

Therefore, we added previous articles reporting on the correlation between tumor localization and histomorphology as a reference. The recurrence rate of the large bile type, represented by the Hilar type, was higher than that of the small bile type, represented by the peripheral type (n=94 [75.2%] vs n=129 [58.6%], p=0.0023). 

We added reference #31-34 as a new reference.

Page 15, line 3-11 in the discussion section.

---

## [Decision Letter · Decision Letter 1]

17 Aug 2020

Efficacy of surgical management for recurrent intrahepatic cholangiocarcinoma: A multi-institutional study by the Okayama study group of HBP surgery

PONE-D-20-19960R1

Dear Dr. Yuzo Umeda,

We’re pleased to inform you that your manuscript has been judged scientifically suitable for publication and will be formally accepted for publication once it meets all outstanding technical requirements.

Kind regards,

Gianfranco D. Alpini

Academic Editor

PLOS ONE

Additional Editor Comments (optional):

Reviewers' comments:

Reviewer's Responses to Questions

**Comments to the Author**

1. If the authors have adequately addressed your comments raised in a previous round of review and you feel that this manuscript is now acceptable for publication, you may indicate that here to bypass the “Comments to the Author” section, enter your conflict of interest statement in the “Confidential to Editor” section, and submit your "Accept" recommendation.

Reviewer #3: All comments have been addressed

2. Is the manuscript technically sound, and do the data support the conclusions?

Reviewer #3: (No Response)

3. Has the statistical analysis been performed appropriately and rigorously? 

Reviewer #3: (No Response)

4. Have the authors made all data underlying the findings in their manuscript fully available?

Reviewer #3: (No Response)

5. Is the manuscript presented in an intelligible fashion and written in standard English?

Reviewer #3: (No Response)

6. Review Comments to the Author

Reviewer #3: (No Response)

7. PLOS authors have the option to publish the peer review history of their article (what does this mean?). If published, this will include your full peer review and any attached files.

Reviewer #3: No

---

## [Editor Report · Acceptance letter]

25 Aug 2020

PONE-D-20-19960R1 

Efficacy of surgical management for recurrent intrahepatic cholangiocarcinoma: A multi-institutional study by the Okayama study group of HBP surgery 

Dear Dr. Umeda:

I'm pleased to inform you that your manuscript has been deemed suitable for publication in PLOS ONE. Congratulations! Your manuscript is now with our production department. 

Kind regards, 

on behalf of

Dr. Gianfranco D. Alpini 

Academic Editor

PLOS ONE